# Farnesoid X receptor activation inhibits TGFBR1/TAK1-mediated vascular inflammation and calcification via miR-135a-5p

Chao Li[1,2], Shijun Zhang[1], Xiaoqing Chen[1], Jingkang Ji[1], Wenqing Yang[1], Ting Gui[1], Zhibo Gai ⓘ [1,2✉] & Yunlun Li ⓘ [1,3✉]

Chronic inflammation plays a crucial role in vascular calcification. However, only a few studies have revealed the mechanisms underlying the development of inflammation under high-phosphate conditions in chronic kidney disease (CKD) patients. Here, we show that inflammation resulting from the activation of the TGFBR1/TAK1 pathway is involved in calcification in CKD rats or osteogenic medium-cultured human aortic smooth muscle cells (HASMCs). Moreover, miR-135a-5p is demonstrated to be a key regulator of the TGFBR1/TAK1 pathway, which has been reported to be decreased in CKD rats. We further reveal that farnesoid X receptor (FXR) activation increases miR-135a-5p expression, thereby inhibiting the activation of the TGFBR1/TAK1 pathway, ultimately resulting in the attenuation of vascular inflammation and calcification in CKD rats. Our findings provide advanced insights into the mechanisms underlying the development of inflammation in vascular calcification, and evidence that FXR activation could serve as a therapeutic strategy for retarding vascular calcification in CKD patients.

[1] Experimental Center, Shandong University of Traditional Chinese Medicine, Jinan 250355, China. [2] Department of Clinical Pharmacology and Toxicology, University Hospital Zurich, Zurich 8032, Switzerland. [3] Affiliated Hospital of Shandong University of Traditional Chinese Medicine, Jinan 250000, China. ✉email: gaizhibo@gamil.com; yunlun.lee@hotmail.com

Vascular calcification is associated with an increased risk of cardiovascular mortality in patients with chronic kidney disease (CKD), independent of classical cardiovascular risk factors[1–3]. It is a prominent feature of atherosclerosis that occurs either at the intima or the tunica media. Medial vessel calcification is one of the most common features in patients with CKD; it occurs mainly in the medial layer of blood vessels, which contains vascular smooth muscle cells (VSMCs), and along the elastic lamellae[4]. Differentiation of VSMCs into an osteoblast-like phenotype is associated with the pathogenesis and pathological process of vascular calcification[5,6]. Recent studies have demonstrated that the expression of osteogenic transcription factors, which regulate the key processes of the differentiation and phenotype determination of osteoblasts, including msh homeobox 2 (Msx2), runt-related transcription factor 2 (Runx2), and osterix, is significantly increased in transformed VSMCs[7]. Furthermore, calcified VSMCs also express some bone-related proteins, including alkaline phosphatase (ALP), osteopontin (OPN), osteocalcin (OCN), and matrix Gla protein (MGP)[3,8]. Impaired calcium and phosphate homeostasis in patients with CKD is considered a major factor for vascular calcification, and phosphate-induced differentiation of VSMCs is decisive for osteoblast differentiation and vascular calcification[9,10]. However, the molecular mechanisms underlying the pathogenesis and potential targeted treatment of vascular calcification in CKD patients have not yet been fully defined.

Human and animal studies have confirmed that aortic inflammation is a critical factor in the pathological process of atherosclerotic calcification[11,12]. Tumor necrosis factor-α (TNF-α) has been reported to stimulate the osteoblastic differentiation of VSMCs by increasing the release and activation of bone morphogenetic protein (BMP)-2[13,14]. Moreover, TNF-α reduces the export of inorganic pyrophosphate (PPi) by activating nuclear factor-κB (NF-κB), thereby decreasing the expression of ankylosis protein homolog (ANKH)[15]. It has also been reported that IL-1β is responsible for osteoblast differentiation and the calcification of VSMCs via the stimulation of tissue-nonspecific alkaline phosphatase (TNAP) activity[16]. Therefore, inflammatory cytokines have been reported to play an important role in promoting vascular calcification. Transforming growth factor-β (TGF-β) is a pleiotropic cytokine that is expressed by all cells and is involved in various cellular functions, especially in inflammatory responses, and cell proliferation and differentiation[17]. After the binding of TGF-β to TGF-β receptor 2 (TGFBR2), TGFBR1 is recruited and phosphorylated, and then, the activities of its downstream molecules SMAD2 and SMAD3 (mothers against decapentaplegic homolog 2 and 3), which are involved in the production of Runx2, are increased[18–20]. In addition, it has been demonstrated that the activation of TGFBR1/2 also acts as an important contributor of NF-kB activation via the TNF receptor-associated factor 6 (TRAF6)/TGF-β-activated kinase 1 (TAK1) pathway[21]. Therefore, whether the TGF-β pathway plays the role of a protagonist in attenuating the formation of vascular calcification by regulating inflammatory responses is yet to be ascertained.

Farnesoid X receptor (FXR) is a member of the nuclear receptor superfamily. It has been demonstrated that FXR is not only expressed in liver and kidney tissues, but also functionally expressed in adult cardiomyocytes and different cell types of the vascular wall[22,23]. After being activated by bile acids, FXR regulates the transcription of target genes involved in bile acids synthesis and metabolism[24]. In addition, FXR also shows an important regulatory effect on glucose metabolism[25], lipid metabolism[26], inflammation[27], and mitochondrial function[28]. A recent study has indicated that FXR activation prevents the osteogenic differentiation of vascular cells in ApoE$^{-/-}$ mice with CKD[29]. However, the roles of FXR in vascular inflammation and the development of calcification and the related underlying mechanisms need to be further studied. We hypothesized whether FXR could postpone the formation of vascular calcification by regulating vascular inflammation.

In recent years, microRNAs (miRNAs) have been considered important regulators of a variety of cell metabolism-related and biological processes; they have been reported to regulate the expression of target genes via the repression of translation or mRNA degradation. MiR-135a-5p has been demonstrated to be involved in a variety of cellular physiological processes. It was reported to show an inhibitory role in several types of cancer including thyroid carcinoma and glioblastoma[30,31]. In addition, miR-135a-5p participates in the neuroprotective effect of hydrogen sulfide on Parkinson's disease[32]. In our previous studies, miRNA transcriptome profiling has been used to screen the crucial regulatory miRNAs that are involved in regulating vascular calcification. In the present study, we aimed to reveal the potential mechanisms underlying the role of inflammatory cytokines in the pathological process of vascular calcification and explore a new therapeutic target for postponing vascular calcification in CKD patients. Moreover, we further characterized the mechanisms underlying the effects of FXR when it is used for the treatment of inflammation and vascular calcification from the perspective of miRNAs.

## Results

**TGFBR1/TAK1 was activated in HASMCs calcification.** Chronic and excessive inflammation plays a crucial role in the development of vascular calcification, but the targets and pathways involved in this process are not yet fully clear. To investigate the regulatory mechanisms associated with inflammation and vascular calcification, we first examined the expression of pro-inflammatory cytokines in HASMCs cultured under high-phosphate conditions. The HASMCs were cultured in osteogenic medium (2.0 mM Pi and 2.7 mM Ca) for 7 days to induce calcification. Culturing in the osteogenic medium increased the calcium accumulation (Fig. 1b) and mineralization in the HASMCs (Fig. 1a). Osteogenic transcription factors and osteogenic markers were also induced by culturing the cells in osteogenic medium, which is consistent with the results of previous studies (Supplementary Fig. 2). Moreover, high levels of pro-inflammatory cytokines and increased intracellular NF-κB contents were also detected in cells cultured in osteogenic medium, compared to those cultured in normal culture medium (Supplementary Fig. 1). These data show that osteogenic medium-induced HASMC calcification resulted in the upregulation of the expression of pro-inflammatory cytokines and increased inflammatory responses. Several studies have indicated that the TGF-β/Smad3 and BMP2 pathways are directly involved in vascular calcification[33,34]; however, to date, no evidence has demonstrated whether the TGFBR1/TAK1 pathway is a key potential target of osteogenic medium-induced HASMC inflammation, which may be also responsible for vascular calcification. We first measured the expression of the TGFBR1/TAK1 pathway in HASMCs cultured in osteogenic medium. Our results showed that the expression of TGFBR1, p-TAK1, TAB1, and p-IκBα increased, in parallel with vascular calcification (Fig. 1c, d). Time course studies during 7 days also showed that the expression of p-TAK1 and NF-κB increased significantly after three days of treating with osteogenic medium, while the calcium content in cells increased significantly after the fifth day ($p < 0.01$) of intervention (Supplementary Fig. 6). Moreover, the further experiments indicated that TGFBR1 is a crucial target of HASMC inflammation, which was indicated by the decrease in the expression of NF-κB and TNF-α after TGFBR1 silencing in HASMCs cultured in

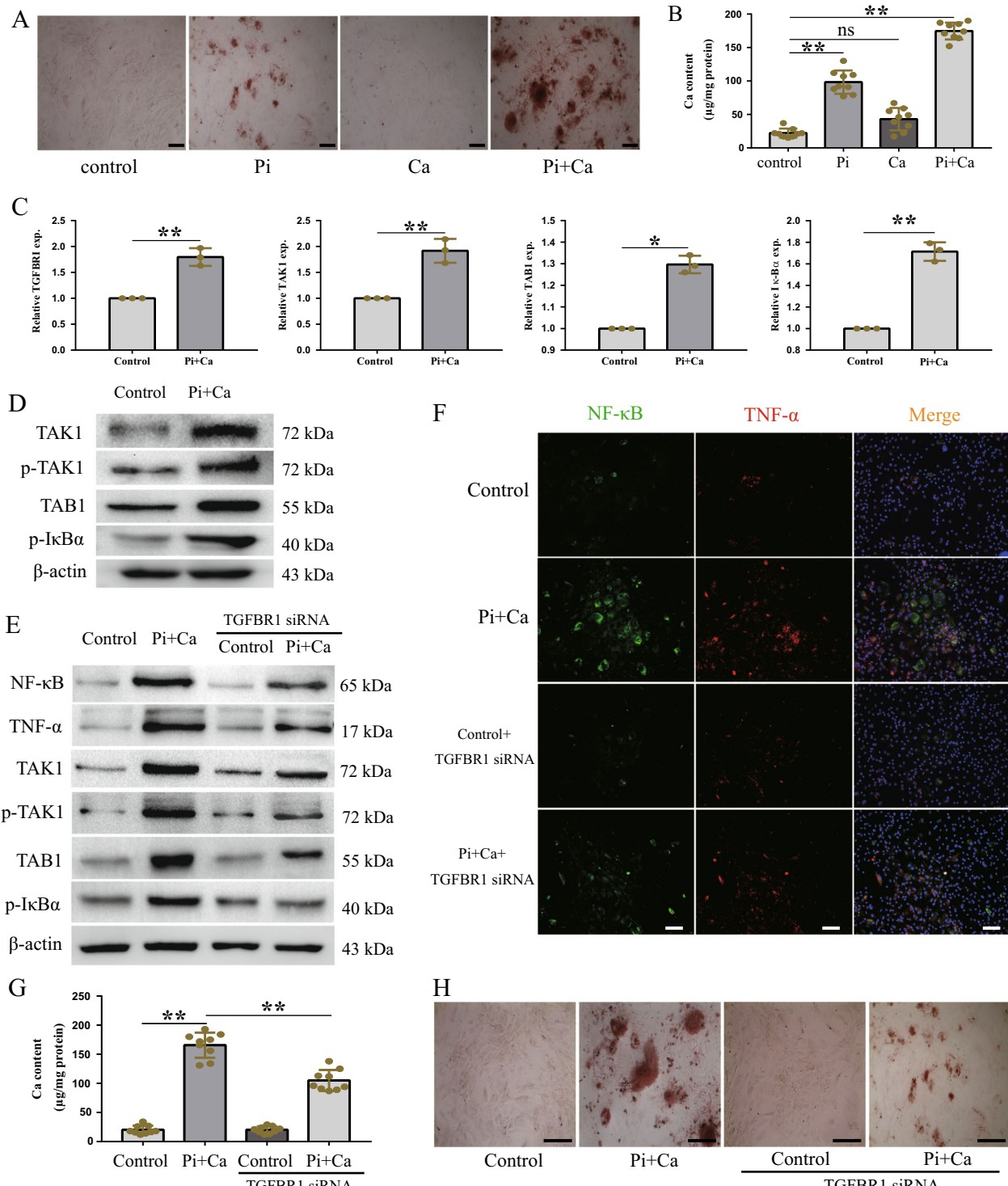

**Fig. 1 TGFBR1/TAK1 pathway was activated by osteogenic medium, resulting in an increased pro-inflammatory cytokines expression and vascular calcification.** HASMCs were cultured with osteogenic medium containing 2.0 mM Pi and 2.7 mM Ca. After 7 days, cells were stained with alizarin red (100×) **a** which identify calcium mineral as red. **b** Ca content assayed by calcium detection assay kit, which was normalized to protein concentration. **c** The mRNA expression of TGFBR1, TAK1, TAB1, and IκBα in osteogenic medium-treated HASMCs. **d** Western blot analysis of the expression of TAK1, p-TAK1, TAB1, and p-IκBα in osteogenic medium-treated HASMCs. **e** Western blot analysis of the expression of NF-κB, TNF-α, TAK1, p-TAK1, TAB1, and p-IκBα in osteogenic medium-treated HASMCs in the absence or presence of TGFBR1 siRNA. **f** Representative immunofluorescence staining of NF-κB (green) and TNF-α (red) in indicated groups (100×). **g** Calcium content of HASMCs in indicated groups. **h** Representative images of alizarin red staining in indicated groups (100×). *P < 0.05, **P < 0.01. ns indicates not significant. Data were pooled as mean ± S.D. (error bars) from three or more independent experiments.

osteogenic medium (Figs. 1f and 1e). The increase in the expression of p-TAK1, TAB1, and p-IκBα in HASMCs cultured in osteogenic medium was also partly reversed after silencing TGFBR1 (Fig. 1e). As expected, osteogenic medium-induced HASMC calcification was alleviated after silencing TGFBR1, as indicated by the results of quantification of the intracellular calcium contents and Alizarin Red S staining (Fig. 1g, h). We also further examined the role of TAK1 in osteogenic medium-cultured HASMCs. As showed in supplementary Fig. 5, TAK1 knockdown attenuated osteogenic medium-induced HASMCs inflammation and calcification. These results, to our knowledge, are the first to indicate that silencing TGFBR1 and TAK1 attenuates osteogenic medium-induced vascular inflammation and calcification, and that the TGFBR1/TAK1 pathway plays a key role in the pathological process of vascular calcification.

**TGFBR1/TAK1 activation-induced vascular calcification in CKD rats**. The rats were fed a high-adenine diet to induce CKD. After four weeks of feeding, high levels of serum creatinine, urea nitrogen, and phosphate were observed in rats fed with the high-adenine diet (HAD group), indicating the impairment of renal function; however, this was not the case in rats fed with a high-phosphate diet (HPD group) (Supplementary Fig. 4). All rats were fed for another four weeks to induce the formation of vascular calcification. Then, Alizarin Red staining and Von Kossa staining (Fig. 2a) of the aortic roots and aortic rings suggested the presence of vascular calcification in rats from the HAD group. Rats from the HAD group, but not the HPD group, showed a increase in the area of calcified lesions (Fig. 2b) and the calcium contents (Fig. 2c) in their aortas, compared to the rats from the control group. In vivo micro-CT imaging also revealed the presence of obvious vascular calcification in rats from the HAD group (Fig. 2d). We further verified the results we found via in vitro experiments. The aortas of rats from the HAD group showed an increased expression of NF-κB and TNF-α, (Fig. 2e, f) compared to the aortic samples from rats in the HPD and control groups. As shown in Fig. 2g, the aortas of rats from the HAD group showed an increased expression of p-TAK1, TAB1, and p-IκBα. These results suggest that inflammation and vascular calcification resulting from the activation of the TGFBR1/TAK1 pathway were observed in rats with CKD.

**FXR activation inhibited TGFBR1/TAK1 and HASMCs inflammation**. Since FXR activation plays a key role in regulating inflammatory responses, whether FXR activation could inhibit TGFBR1/TAK1 signaling and thus, alleviate the osteogenic medium-induced HASMC inflammation was examined. We first tested the expression of the TGFBR1/TAK1 pathway in HASMCs cultured with osteogenic medium and obeticholic acid (OCA, a FXR agonist). Our results showed that the TGFBR1/TAK1 pathway was activated in HASMCs cultured in osteogenic medium; this activation was inhibited by OCA (Fig. 3b). As expected, the increase in the expression of NF-κB and TNF-α in the HASMCs cultured in osteogenic medium was reduced following OCA supplementation (Fig. 3b, c). OCA also decreased the intracellular NF-κB contents, which increased in the HASMCs cultured in osteogenic medium (Fig. 3d). These data suggest that FXR activation inhibits the TGFBR1/TAK1 pathway, thus reducing the expression of pro-inflammatory cytokines in the HASMCs cultured in osteogenic medium.

**FXR activation retarded vascular calcification formation**. Our study has demonstrated that the inflammation mediated by the TGFBR1/TAK1 pathway was involved in the pathological process of vascular calcification, which was also inhibited by OCA.

Therefore, we hypothesize that FXR activation could attenuate the formation of calcification by inhibiting inflammation. We further measured the osteogenic differentiation of HASMCs cultured in osteogenic medium after their treatment with OCA. As shown in Fig. 3e, f, OCA reduced the osteogenic medium-induced HASMC calcification and decreased the intracellular calcium contents. OCA also inhibited the expression levels of Runx2 and ALP, which were upregulated following the culturing of the cells in the osteogenic medium (Fig. 3g). Consistent with the in vitro results, simultaneous feeding of OCA and the high-adenine diet (OCA group) postponed the formation of vascular calcification in rats with CKD, but OCA treatment after feeding the rats with the high-adenine diet for four weeks (OCA-T group) had no effects with regards to the reversal of vascular calcification (Fig. 4a). Micro-CT imaging of the rats also showed that the formation of vascular calcification was postponed in rats from the OCA group, but not in those from the OCA-T group (Fig. 4b). The results of the analyses of the calcified lesion areas (Fig. 4c) and aortic calcium contents (Fig. 4d) were also consistent with the staining results. However, OCA had no effect on repairing kidney failure and regulating high-phosphate condition (Supplementary Fig. 3). These data show that OCA-mediated FXR activation alleviates the formation of vascular calcification but does not reverse the calcification in areas that have already been calcified in CKD rats. Furthermore, our study showed that OCA treatment reduced vascular inflammation in CKD rats (Fig. 4e, f). These data suggest that FXR activation inhibits TGFBR1/TAK1-mediated inflammation and retards the formation of vascular calcification.

**MiR-135a-5p was the target of FXR to attenuate calcification**. Based on our previous study, we selected some TGF-β pathway-associated miRNAs that were differentially expressed in HASMCs cultured in osteogenic medium, as shown in Fig. 5a. Four of the miRNAs whose expressions showed significant differences were selected for validation ($p < 0.05$). The results showed that the miR-135a-5p expression was decreased in HASMCs cultured in osteogenic medium; this decrease was reversed by OCA (Fig. 5b, d). The expression of miR-135a-5p in the vascular cells of CKD rats was consistent with the miR-135a-5p expression results obtained in the in vitro analysis (Fig. 5c). We hypothesized that miR-135a-5p is a critical target of OCA in the regulation of inflammation. Therefore, we measured the intracellular NF-κB content and pro-inflammatory cytokine expression in miR-135a-5p inhibitor-treated HASMCs in the absence or presence of OCA. The results showed that OCA reduced the intracellular NF-κB content and the expression of NF-κB and TNF-α in the HASMCs cultured in osteogenic medium; this reduction was partly abolished by treatment with the miR-135a-5p inhibitor (Fig. 5e, f). We further examined the intercellular calcium contents. As shown in Fig. 5g, OCA reduced the increase of the intercellular calcium contents in the HASMCs cultured in osteogenic medium; this reduction was partly reversed by the miR-135a-5p inhibitor. The results of the Alizarin Red staining analysis also showed that OCA inhibited the osteogenic medium-induced osteogenic differentiation of HASMCs, and the miR-135a-5p inhibitor suppressed this effect (Fig. 5h). These data suggest that FXR activation upregulates the miR-135a-5p expression, thereby reducing the osteogenic medium-induced inflammation, and then inhibits the osteogenic differentiation of HASMCs.

**FXR activation increased miR-135a-5p to inhibit TGFBR1/TAK1**. To further explore the potential targets of miR-135a-5p, we used bioinformatics target prediction software (TargetScan Release 4.2, miRSystem, and RNAhybrid) to analyze the potential targets of miR-135a-5p in the TGF-β pathway. It was found that

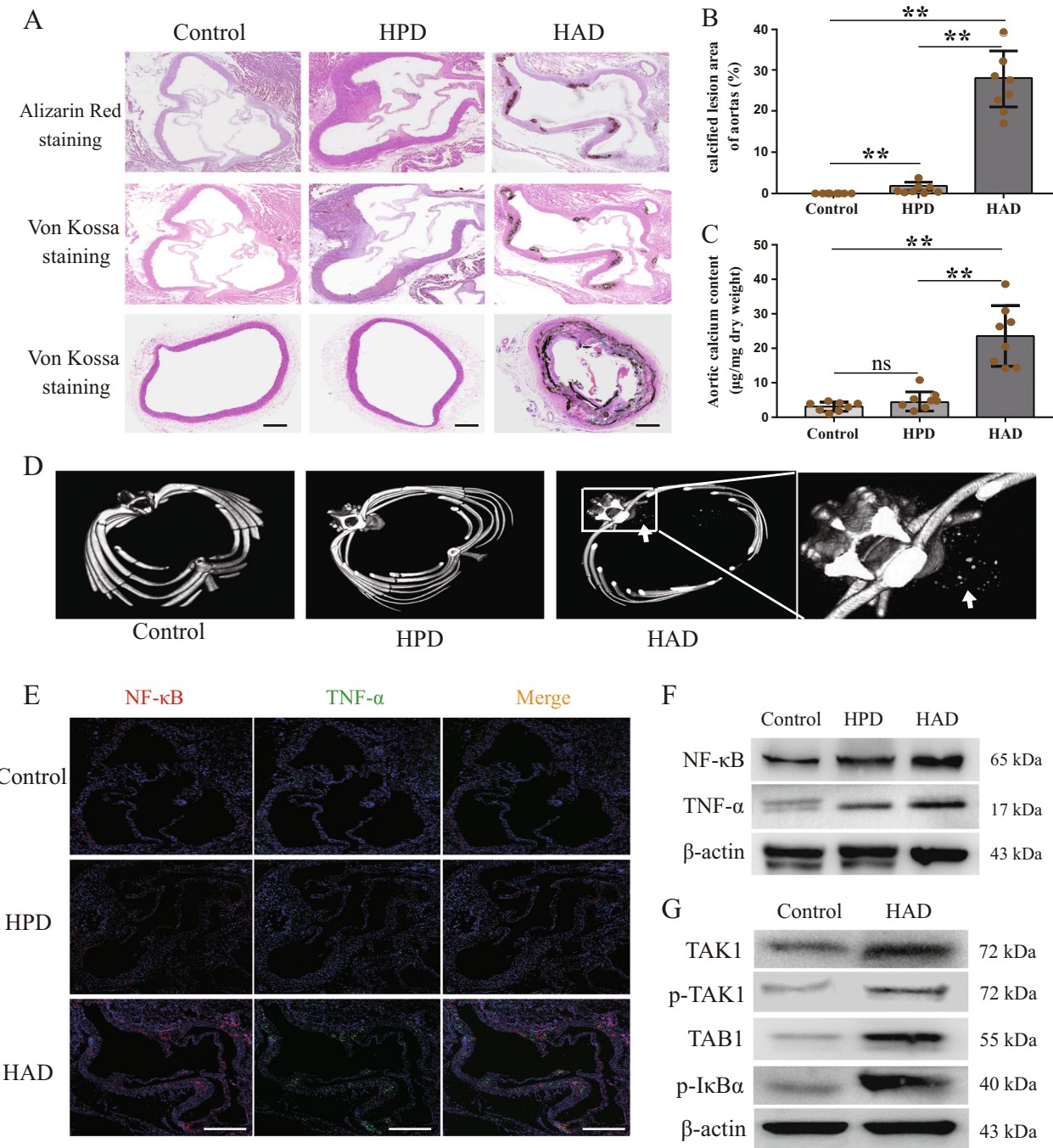

**Fig. 2 Activation TGFBR1/TAK1 pathway-mediated inflammation and vascular calcification were observed in rats with CKD.** Wistar rats were fed either chow diet or high-phosphate diet in the absence or presence of 0.75% adenine to induce chronic kidney disease for 4 weeks, all rats were fed for another 4 weeks to form vascular calcification (n = 8 per group). **a** Representative images of alizarin red staining and von Kossa staining of aortic roots or aortic ring in indicated groups (scale bar, 500 μm). **b** Calcified lesion areas of aortic in indicated groups. **c** Aortic calcium content analysis in indicated groups. **d** Representative images of micro-CT, arrows indicate calcified nodules. **e** Representative immunofluorescence staining of NF-κB (red) and TNF-α (green) of aortic roots in indicated groups (scale bar, 500 μm). **f** Western blot analysis of NF-κB and TNF-α expression in aortas in indicated groups. **g** Western blot analysis of the expression of TAK1, p-TAK1, TAB1, and p-IκBα in high-adenine diet-fed rats (HAD group, n = 8 per group). **P < 0.01, ns indicates not significant.

the 3′-UTR of TGFBR1 contains a putative binding site for miR-135a-5p (Fig. 6a). To determine whether TGFBR1 is a direct target of miR-135a-5p, miR-135a-5p inhibitor and mimics were transfected into HASMCs. The results showed that the miR-135a-5p mimics inhibited the expression of TGFBR1, but there was no difference in the TGFBR1 expression after the cells were transfected with the miR-135a-5p inhibitor, compared to the case for

the cells in the control group (Fig. 6b). HASMCs transfected with the miR-135a-5p mimics were cultured in the presence of TGF-β1, and the expression levels of some markers present downstream of the TGFBR1/TAK1 pathway were measured to further confirm the miR-135a-5p-mediated inhibition of the TGFBR1 expression. As shown in Fig. 6c, d, TGF-β1 increased the expression of p-TAK1, TAB1, and IκBα; however, this effect was

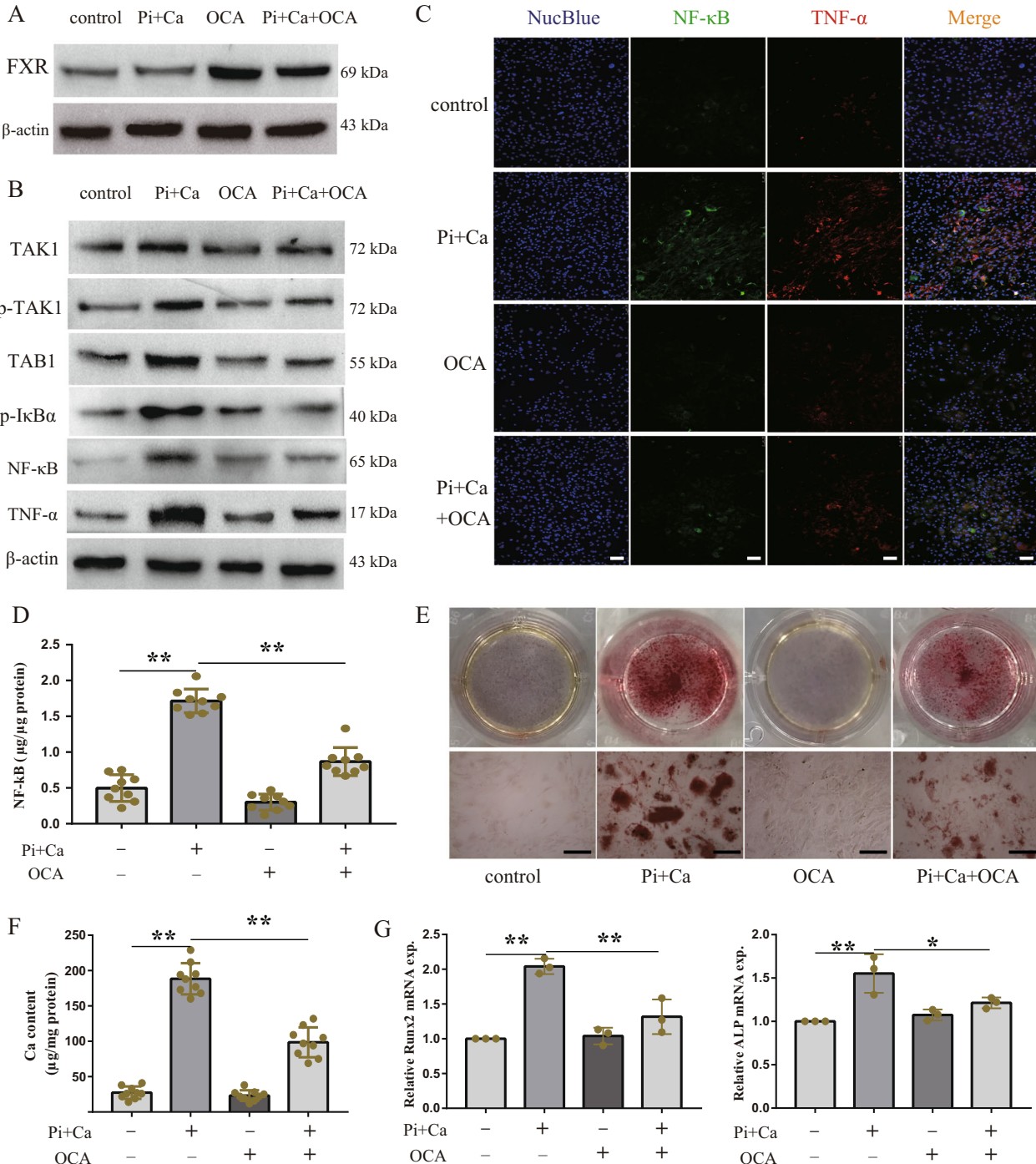

**Fig. 3 FXR activation inhibited TGFBR1/TAK1 pathways to attenuate osteogenic medium-induced HASMCs inflammation and calcification.** HASMCs were cultured with osteogenic medium in the absence or presence of 3 μM OCA (FXR ligand) for 7 days. **a** Western blot results showed that FXR was activated by OCA. **b** Western blot analysis of the expression of TAK1, p-TAK1, TAB1, p-IκBα, NF-κB, and TNF-α in indicated groups. **c** Representative immunofluorescence staining of NF-κB (green) and TNF-α (red) in indicated groups (100×). **d** Intercellular NF-κB content of HASMCs in indicated groups. **e** Representative images of alizarin red staining in indicated groups (100×). **f** Calcium content of HASMCs in indicated groups. **g** The mRNA expression of Runx2 and ALP in indicated groups. *P < 0.05, **P < 0.01. Data were pooled as mean ± S.D. (error bars) from three or more independent experiments.

reversed after the transfection of the cells with the miR-135a-5p mimics. We further examined the expression of p-TAK1, TAB1, and p-IκBα in OCA-treated HASMCs that were transfected with the miR-135a-5p inhibitor at the same time. As shown in Fig. 6e, OCA decreased the expression levels of p-TAK1, TAB1, and IκBα, which were higher in the HASMCs cultured in osteogenic medium. However, the miR-135a-5p inhibitor weakened the effect of OCA. These data suggest that TGFBR1 is a direct target

of miR-135a-5p and that OCA-mediated FXR activation upregulates the miR-135a-5p expression, thereby inhibiting the TGFBR1/TAK1 pathway.

**FXR attenuated vascular calcification via miR-135a-5p/TGFBR1/TAK1.** To further confirm that miR-135a-5p is a key target of FXR activation, which results in the attenuation of

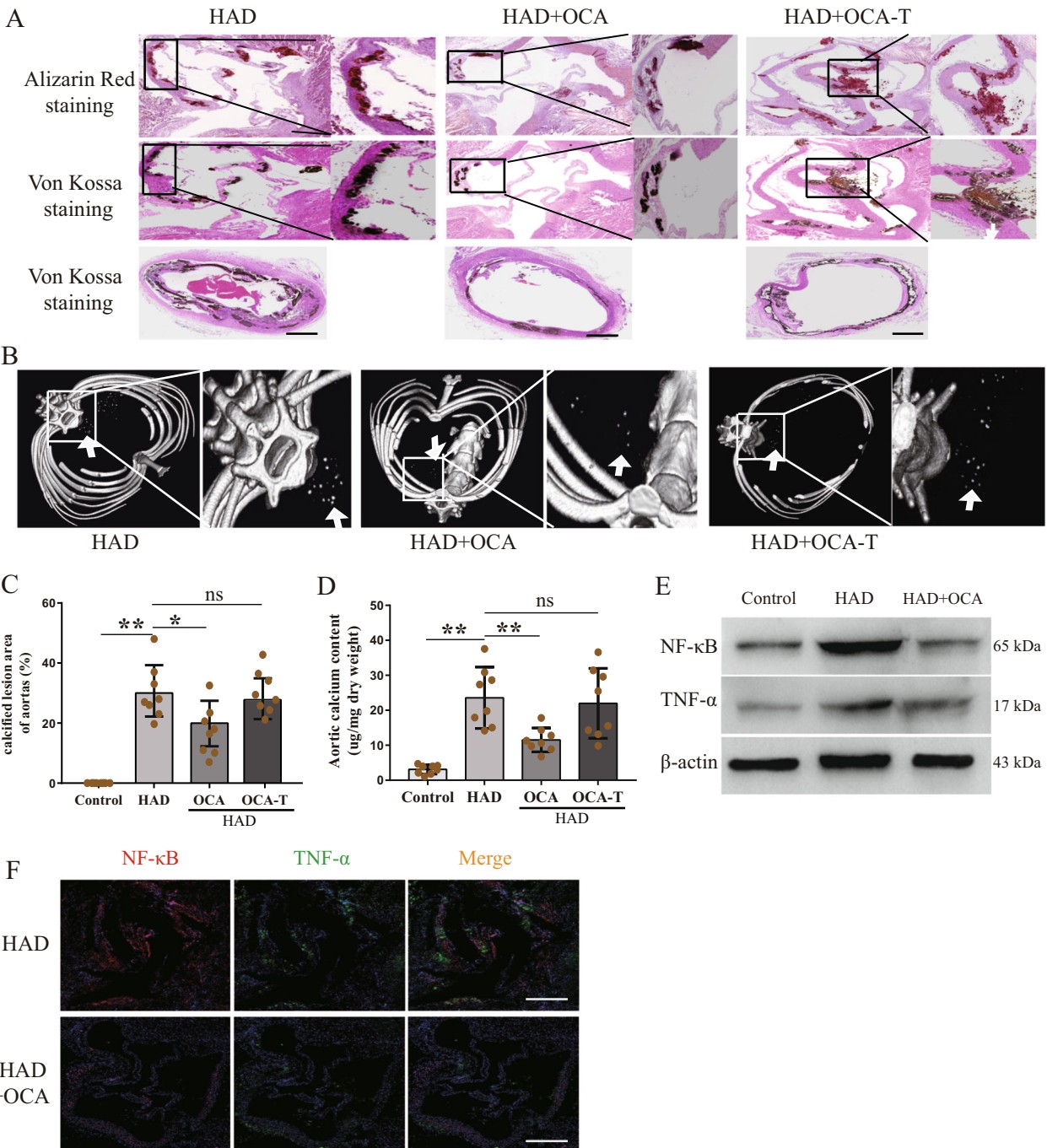

**Fig. 4 FXR activation retarded vascular calcification formation and attenuated vascular inflammation in CKD rats.** FXR ligand (OCA) was administered to rats by gavage at a dose of 10 mg/kg/day. In order to detect the effect of OCA, one group of rats was given OCA in parallel with feeding high-adenine diet (named OCA group), while the other group was given OCA by 4 weeks after feeding high-adenine diet (named OCA-T group). **a** Representative images of alizarin red staining and von Kossa staining of aortic roots or aortic ring in indicated groups (scale bar, 500 μm). **b** Representative images of micro-CT in indicated groups, arrows indicate calcified nodules. **c, d** Calcified lesion areas of aortic (**c**) and aortic calcium content (**d**) in indicated groups. **e** Western blot analysis of NF-κB and TNF-α expression in aortas in indicated groups. **f** Representative immunofluorescence staining of NF-κB (red) and TNF-α (green) of aortic roots in indicated groups (scale bar, 500 μm). $n = 8$ per group, *$P < 0.05$, **$P < 0.01$, ns indicates not significant.

vascular inflammation and calcification in CKD rats, we injected antagomiR-135a-5p into rats through the vena caudalis for three consecutive days before inducing CKD. Since antagomirs specific and complementary to their mature target miRNA, interfere with its function and are proved to be efficient in many tissues, we used antagomiR-135a-5p to inhibit miR-135a-5p in animal experiments. As shown in Fig. 7a, TGFBR1 expression was increased in the aortas of rats that were injected with

antagomiR-135a-5p. AntagomiR-135a-5p-treated rats developed vascular calcification, but the degrees of calcification and calcified lesion areas in antagomiR-135a-5p-treated rats were lesser than those in case of rats with CKD that were not treated with antagomiR-135a-5p (Fig. 7b–d). These data indicated that miR-135a-5p is partly involved in the pathological mechanism of CKD-induced vascular calcification. We further measured the vascular calcification in antagomiR-135a-5p-treated rats that

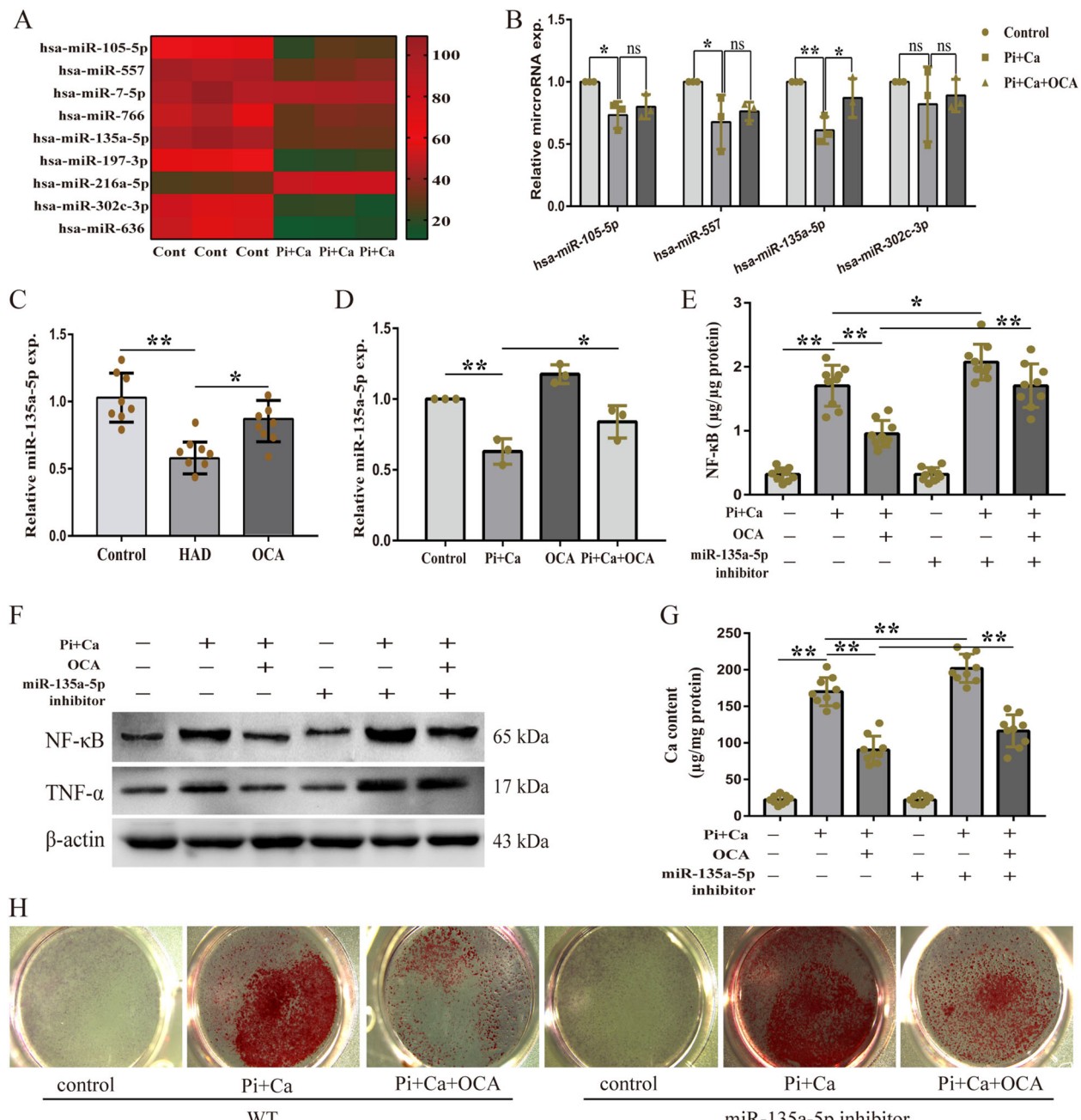

**Fig. 5 FXR activation attenuated osteogenic medium-induced HAMSCs inflammation and calcification via increasing miR-135a-5p. a** TGF-β pathway-associated microRNAs differentially expressed in osteogenic medium-cultured HASMCs assayed by high-throughput sequencing. **b** The expression of microRNA in osteogenic medium-cultured HASMCs in the absence or presence of OCA. **c** MiR-135a-5p expression of aortas in indicated groups (*n* = 8 per group). **d** MiR-135a-5p expression of HASMCs in indicated groups. **e** Intercellular NF-κB content of HASMCs in indicated groups. **f** Western blot analysis of NF-κB and TNF-α expression in HASMCs in indicated groups. **g** Calcium content of HASMCs in indicated groups. **h** Representative images of alizarin red staining in indicated groups. *P < 0.05, **P < 0.01, ns indicates not significant. Data were pooled as mean ± S.D. (error bars) from three or more independent experiments.

were also administered in OCA. Alizarin Red staining analysis, examination of the calcified lesion areas, and quantification of the aortic calcium contents showed that the OCA-mediated retardation of vascular calcification was weakened by antagomiR-135a-5p (Fig. 7b–d). The results of the micro-CT analysis also showed that antagomiR-135a-5p reduced the OCA-mediated retardation of the formation of vascular calcification (Fig. 7e). With regards to vascular inflammation, as expected, antagomiR-135a-5p partly abolished the OCA-mediated

inhibition of the expression of inflammatory cytokines (Fig. 7f). In addition, OCA inhibited the expression of p-TAK1, TAB1, and p-IκBα, which are present downstream of miR-135a-5p/TGFBR1 signaling; this inhibition was also attenuated by antagomiR-135a-5p (Fig. 7f, g). These data suggest that miR-135a-5p is a key target of FXR activation, which causes the inhibition of the TGFBR1/TAK1 pathway, thereby resulting in the attenuation of vascular inflammation and calcification in CKD rats.

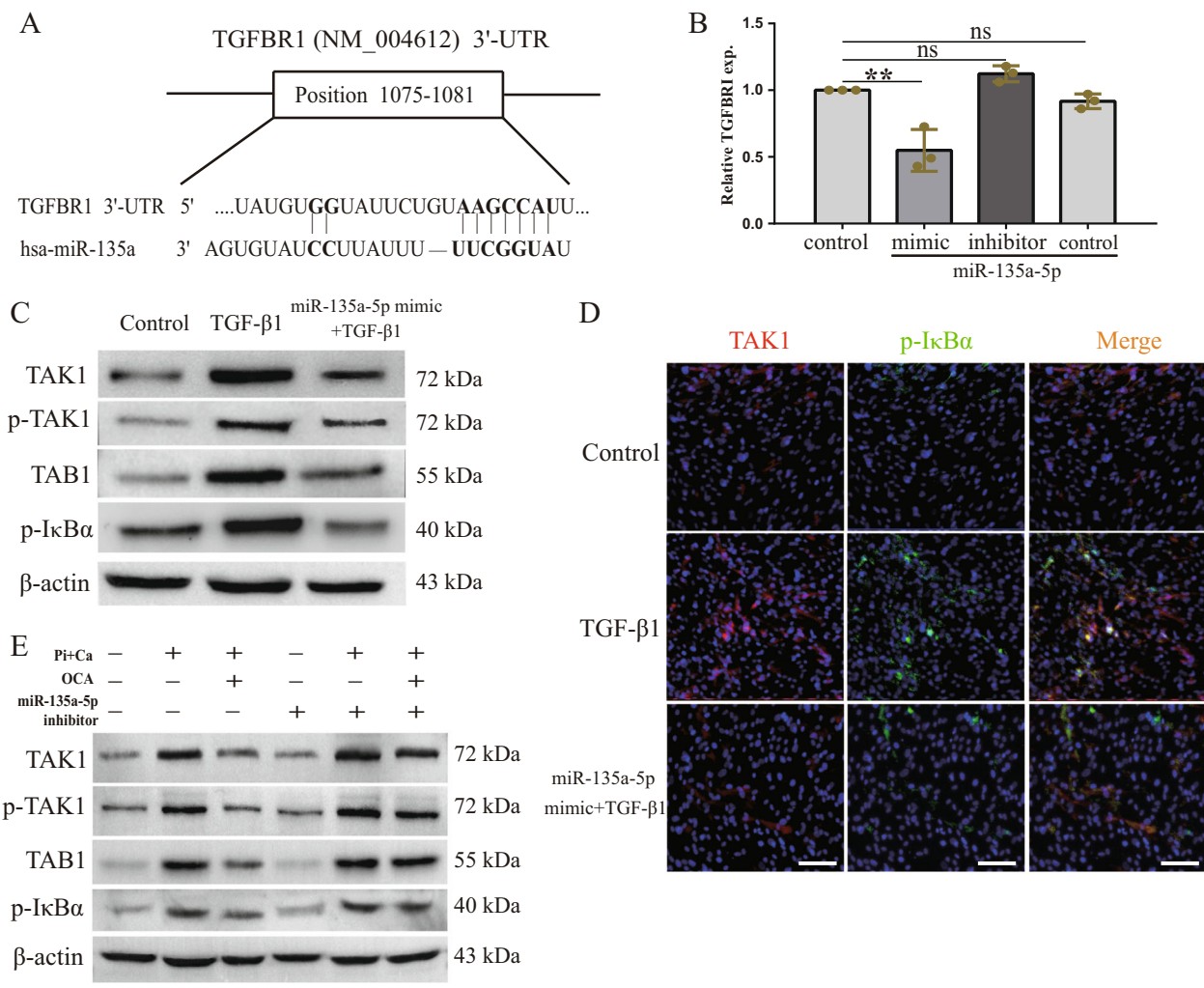

**Fig. 6 FXR activation inhibited osteogenic medium-induced TGFBR1/TAK1 pathway activation via increasing miR-135a-5p expression. a** Potential sequences of the miR-135a-5p binding sites within the 3'-UTR of TGFBR1 was predicted by TargetScan Release 4.2 and RNAhybird. **b** The mRNA expression of TGFBR1 after overexpressing or inhibiting miR-135a-5p. **c** Western blot analysis of p-TAK1, TAB1, and p-IκBα in TGF-β1-cultured HASMCs in the absence or presence of miR-135a-5p mimic. **d** Representative immunofluorescence staining of TAKI (red) and p-IκBα (green) in indicated groups (100×). **e** Western blot analysis of TAK1, p-TAK1, TAB1, and p-IκBα in indicated groups. **P < 0.01, ns indicates not significant. Data were pooled as mean ± S.D. (error bars) from three or more independent experiments.

## Discussion

Vascular calcification is a major risk factor for cardiovascular damage in patients with CKD; chronic and excessive inflammation plays a crucial role in vascular calcification. However, studies revealing the mechanisms underlying the development of inflammation under high-phosphate conditions in CKD patients are rare. Our present study has shown that: (i) osteogenic medium-induced vascular calcification involves inflammatory responses, (ii) HASMC inflammation mediated by the TGFBR1/TAK1 pathway is involved in the pathological process of vascular calcification, (iii) the FXR agonist OCA retards the formation of vascular calcification by reducing TGFBR1/TAK1-mediated inflammation, and (iv) FXR activation upregulates the miR-135a-5p level, thereby inhibiting the activation of the TGFBR1/TAK1 pathway in CKD rats; this results in a decrease of NF-κB expression, and eventually, the attenuation of vascular inflammation and the formation of calcification. Our results suggest that TGFBR1/TAK1 pathway-mediated vascular inflammation plays a critical role in the pathological process of vascular calcification and that FXR activation prevents the development of CKD-induced vascular inflammation and calcification *via* the modulation of the miR-135a-5p/TGFBR1/TAK1 pathway; this

underpins the importance of FXR as a new potential target for the treatment of vascular calcification in patients with CKD.

A comprehensive meta-analysis of prospective studies reporting cardiovascular end-points and calcifications has revealed that the odds ratio for cardiovascular mortality in patients with vascular calcification was 3.94 (95% CI, 2.39–6.50), suggesting that the high prevalence of vascular calcification in patients is associated with an increased risk for adverse cardiovascular events[2]. The pathogenesis of vascular calcification is multifactorial; a high level of phosphate is considered a major factor that activates the expression of osteogenic transcription factors, including Msx2, Runx2, and osterix, and then induces vascular calcification[35]. Our present study showed that osteogenic medium not only induced HASMC calcification, but also increased the expression of pro-inflammatory cytokines. In vivo experiments also showed that the increase in the expression of pro-inflammatory cytokines occurred in parallel with vascular calcification in rats with CKD. An increasing number of studies have demonstrated that pro-inflammatory cytokines are involved in the activation of some osteogenic transcription factors and expression of some bone-related proteins[11,13–15]; however, the exact pathway of inflammatory response activation in vascular calcification has not yet

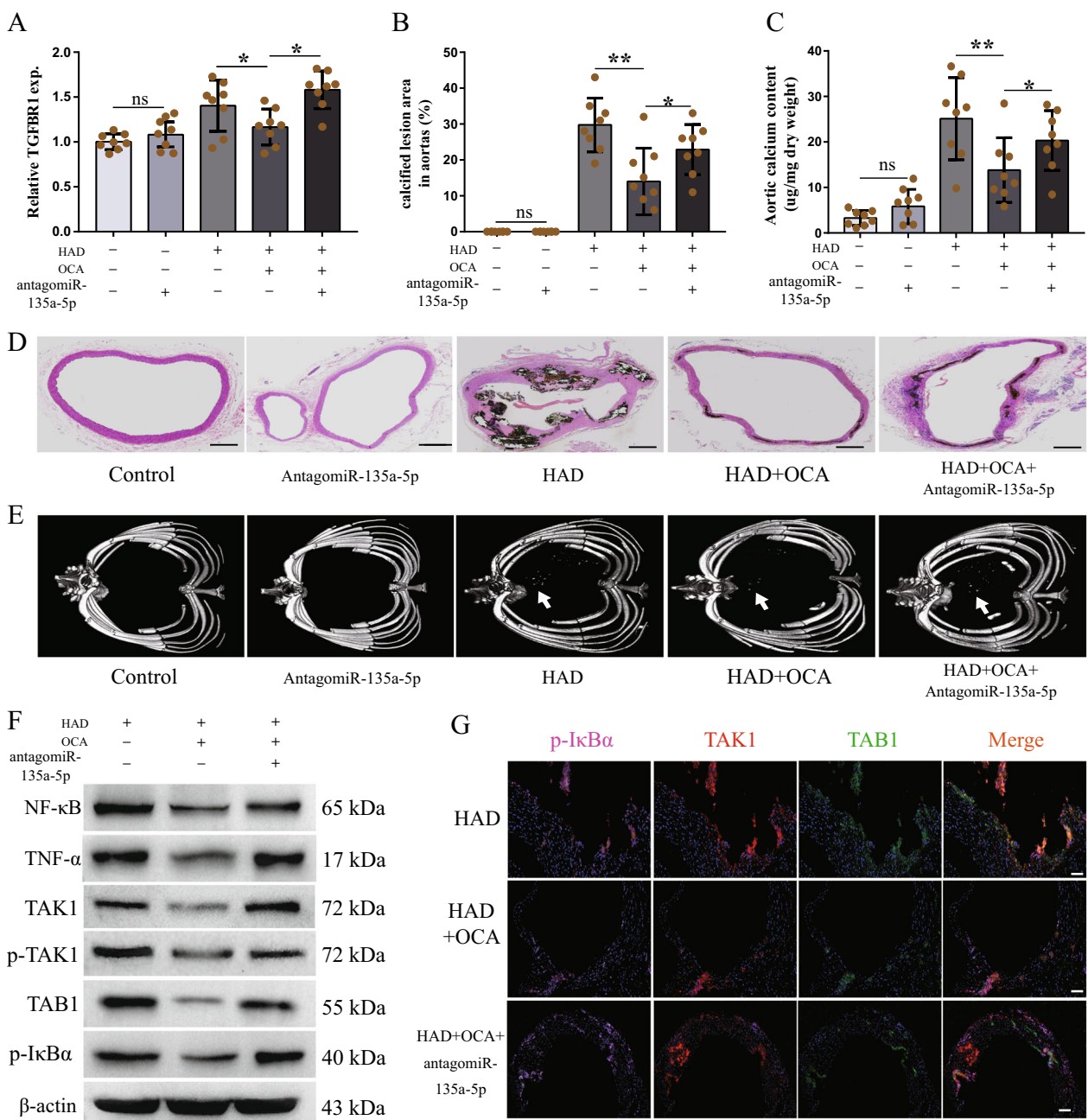

**Fig. 7 AntagomiR-135a-5p weakened the effect of OCA on mitigating vascular inflammation and calcification in CKD rats.** To inhibit miR135a-5p, rats were injected with antagomiR135a-5p (80 μg/g body weight) through vena caudalis for three consecutive days before inducing chronic kidney disease (*n* = 8 per group). **a** The mRNA expression of TGFBR1 of aortas in indicated groups. **b**, **c** Calcified lesion areas of aortic (**b**) and aortic calcium content (**c**) in indicated groups. **d** Representative images of von Kossa staining of aortic ring in indicated groups (scale bar, 500 μm). **e** Representative images of micro-CT in indicated groups, arrows indicate calcified nodules. **f** Western blot analysis of NF-κB, TNF-α, TAK1, p-TAK1, TAB1, and p-IκBα of aortas in indicated groups. **g** Representative immunofluorescence staining of p-IκBα (pink), TAK1 (red) and TAB1 (green) of aortic rings in indicated groups (scale bar, 500μm). *P < 0.05, **P < 0.01, ns indicates not significant.

been fully clarified. Based on our present results, we put forward an assumption that the activation of inflammatory cytokines plays a key role in the formation of vascular calcification, and inflammation could be a key therapeutic target of vascular calcification. Therefore, our future studies will focus on the mechanisms underlying inflammatory response activation in vascular calcification, and on ascertaining whether inflammation is a regulatory target for vascular calcification.

Our present study showed that the TGFBR1/TAK1 pathway was activated in CKD rats and HASMCs cultured in osteogenic

medium. The expression of TAB1, p-IκBα, NF-κB, and TNF-α, which are involved in processes occurring downstream of the TGFBR1/TAK1 pathway, also increased, in parallel with vascular calcification. After silencing TGFBR1, the TGFBR1/TAK1 pathway-mediated inflammation and calcification were found to have decreased. It is known that TGF-β is expressed by all cells and that TGF-β signaling plays a crucial role in regulating normal inflammatory responses[36]. Studies have demonstrated that TGF-β1 expression is increased by phosphate in VSMCs[37,38]. TGF-β1 is the ligand of TGFBR2, and when TGF-β1 bind to TGFBR2;

then, TGFBR1 is recruited and phosphorylated, which results in the phosphorylation of SMAD2/3 and the regulation of the transcription of the target genes of TGF-β[18,19]. On the other hand, activated TGFBR1 can also cause the K63-polyubiquitylation of TAK1 (non-canonical TGF-β signaling), which has widely been considered a pivotal regulator of the expression of pro-inflammatory cytokines including NF-κB and TNF-α[21,39,40]. Therefore to our knowledge, our study is the first to discover that the TGFBR1/TAK1 pathway-mediated HASMC inflammation is involved in the pathological process of vascular calcification; the inhibition of this inflammation could serve as a new therapeutic target for retarding vascular calcification. Moreover, TAK1 also plays a key role in MAPKs activation that is involved in JNK and p38 mainly. Some studies have demonstrated that p38 MAPK activation is involved in the Pi-induced smooth muscle cells inflammation and calcification, and inhibition of p38 MAPK decreases the smooth muscle cells calcification[41–43]. These studies also provide evidence that TGFBR1/TAK1 could be a good target for reducing vascular calcification.

It has been demonstrated that FXR activation plays a key role in regulating inflammatory responses[44]. It has been reported that FXR$^{-/-}$ mice show a higher TNF-α mRNA level than wild-type mice[45]. Moreover, a recent experimental study has demonstrated that FXR activation inhibited IL-1β-induced NF-κB activation in VSMCs and then reduced the expression of iNOS and COX-2, which contribute to vascular inflammation, VSMC migration, and formation of atherosclerotic lesions[46]. However, no studies have revealed the effects of FXR activation on vascular inflammation and calcification in CKD rats and the related underlying mechanisms. Our study showed that FXR activation inhibited the TGFBR1/TAK1 pathway, which was activated in CKD rats and HASMCs cultured in osteogenic medium, causing a reduction in the expression of pro-inflammatory cytokines and postponing the formation of vascular calcification. However, OCA did not show the therapeutic effect on CKD. The results of one previous study were found to be similar to those of our study; this previous study has reported that lipid accumulation exists in the vascular calcification associated with atherosclerosis, and FXR activation directly ameliorated vascular calcification by activating the c-Jun N-terminal kinase in 5/6 nephrectomized ApoE$^{-/-}$mice[29]. FXR activation can regulate bile acid synthesis and cholesterol metabolism, which is closely related to lipid accumulation and atherosclerosis[47,48]. However, the mechanisms underlying the FXR activation- mediated inhibition of the TGFBR1/TAK1 pathway and the alleviation of vascular inflammation and calcification are still unclear.

In our previous studies, we have screened the expression of miRNAs in VSMCs showing osteoblastic differentiation by high-throughput sequencing and established the presence of different miRNA expression profiles. MiRNAs are a class of endogenous small non-coding RNAs comprising 18–25 nucleotides; they are responsible for regulating the expression of multiple protein-encoding genes in various biological processes[49]. Previous studies have shown that miRNAs play a critical role in the development of vascular calcification, inflammation, and vascular dysfunction[50,51]. MiR-125b has been reported to inhibit SP7 expression, resulting in the osteogenic differentiation of VSMCs[52]. MiR-204 has been shown to attenuate medial artery calcification by downregulating its target gene Runx2[53]. The expression levels of different miRNAs involved in TGFBR1 signaling have been verified in HASMCs cultured in osteogenic medium and OCA. Our results showed that the miR-135a-5p expression was decreased in HASMCs cultured in osteogenic medium and CKD rats; however, this decrease was reversed after treatment with OCA. MiR-135a-5p has been demonstrated to be involved in the hypoxia-induced proliferation

of pulmonary artery smooth muscle cells[54]. MiR-135a-5p has also been reported to suppress 3T3-L1 preadipocyte differentiation and adipogenesis via the activation of canonical Wnt/b-catenin signaling[55]. However, the roles of miR-135a-5p in vascular inflammation and calcification and the related underlying mechanisms have rarely been reported. After the inhibition of miR-135a-5p, the FXR activation-mediated alleviation of inflammation and osteoblast differentiation was reduced. Therefore, miR-135a-5p is partly responsible for the FXR activation-mediated attenuation of vascular inflammation and calcification.

To further confirm if TGFBR1 signaling is a potential target of miR-135a-5p, various bioinformatics target prediction software were used to analyze the potential targets of miR-135a-5p in the TGF-β pathway. We found that in the 3′-UTR of the TGFBR1 gene, there is a potential site for the binding of the miR-135a-5p sequence. In our present study, transfection with the miR-135a-5p mimic inhibited the TGF-β1-activated TGFBR1/TAK1 pathway, and transfection with the miR-135a-5p inhibitor attenuated the OCA-mediated inhibition of the TGFBR1/TAK1 pathway; this pathway was found to be activated by the osteogenic medium. In vivo studies also demonstrated that OCA inhibited the activation of the TGFBR1/TAK1 pathway, thereby postponing the formation of vascular calcification in CKD rats; this effect was weakened by antagomiR-135a-5p. Therefore, miR-135a-5p may be also a key target for vascular calcification, since increased miR-135a-5p expression by FXR activation reduces the formation of vascular calcification. However, miR-135a-5p has no effect when vascular calcification has been formed, so it could not reverse calcified areas that have been calcified according to the CKD rat experiment. Although the miR-135a-5p inhibitor or antagomiR-135a-5p inhibited the activation of the TGFBR1/TAK1 pathway under high-phosphate conditions in CKD rats, no effects of these substances were observed in normal cells or rats.

In summary, to our knowledge, the present study has revealed a novel mechanism underlying the development of inflammation in vascular calcification. As shown in Fig. 8, our findings suggest that the activation of the TGFBR1/TAK1 pathway is involved in the pathological process of vascular calcification, and that miR-135a-5p is a key regulator of the TGFBR1/TAK1 pathway. FXR activation increases the miR-135a-5p expression, thereby inhibiting the activation of the TGFBR1/TAK1 pathway, and resulting in the attenuation of vascular inflammation and calcification in CKD rats without repairing kidney failure. Therefore, our present data provide evidence that FXR could serve as a new therapeutic target for retarding the formation of vascular calcification in CKD patients. However, further clinical studies are required to verify the effects of FXR activation; this will be the focus of our future studies.

## Methods

**Cell culture and treatment.** Human aortic smooth muscle cells (HASMCs) were purchased from Sciencell (#6110, San Diego, California, USA); they were cultured in smooth muscle cell medium containing 2% fetal bovine serum (FBS), 1% penicillin/streptomycin solution (P/S), and 1% smooth muscle cell growth supplement (SMCGS) (Sciencell, San Diego, California, USA) at 37 °C and 5% $CO_2$ conditions in a humidified incubator. $CaCl_2$ and $NaH_2PO_4$ were added to the culture medium to prepare an osteogenic medium with final Pi and Ca concentrations of 2.0 mM and 2.7 mM, respectively. The HASMCs were cultured in the osteogenic medium for 7 days to induce calcification, which was identified by Alizarin Red S staining and calcium content analysis. Obeticholic acid (OCA/INT-747) was added to the osteogenic medium (final concentration of 3 μM), and the medium was changed every 2–3 days.

**Animal experiments.** The animal experiments were performed in accordance with the Guide for the Care and Use of Laboratory Animals (published by the US National Institutes of Health) and were approved by the Institutional Animal Care and Research Advisory Committee of the Shandong University of Traditional Chinese Medicine. As shown in Fig. 9, Wistar rats (10-week-old rats, male and female in half) were fed either normal chow or a high-phosphate diet (containing

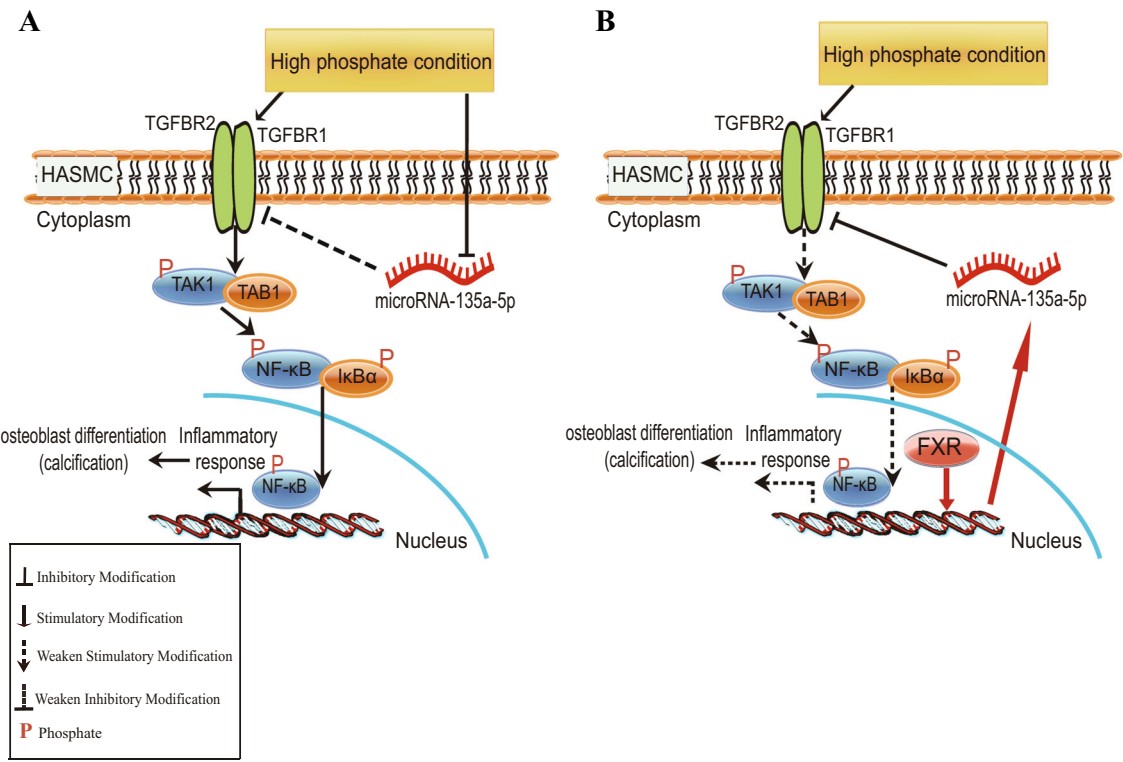

**Fig. 8 Schematic model for FXR activation attenuating TGFBR1/TAK1 pathway-mediated inflammation and calcification through increasing microRNA-135a-5p expression in osteogenic medium-cultured HASMCs. a** TGFBR1 is activated in osteogenic medium-cultured HASMCs, and then activates TAK1, which binds with TAB1 and TAB2 to form a stable complex at the N-terminal kinase domain and the C-terminal region of TAK1. NF-κB pathway is the downstream of TAK1, which is reported to stimulate the expression of osteogenic transcription factor and osteogenic marker, resulting in calcification formation of HASMCs. MicroRNA-135a-5p is key regulator to inhibited TGFBR1, which is suppressed in osteogenic medium-cultured HASMCs. **b** FXR activation increases microRNA-135a-5p expression to inhibit TGFBR1/TAK1 pathway and further attenuates osteogenic medium-induced HASMCs inflammation and calcification.

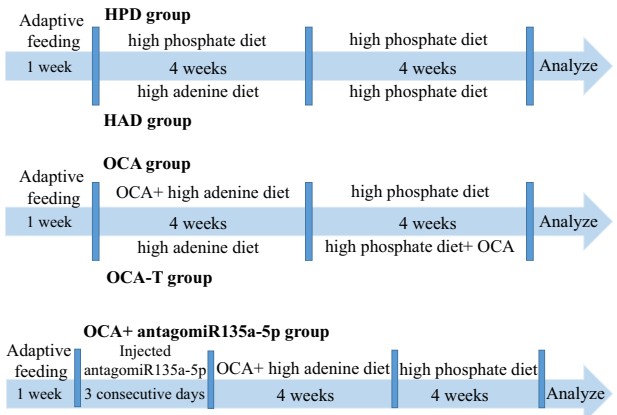

**Fig. 9 Schematic diagram of rat CKD model induction and drug intervention.** All rats were accepted adaptive feeding for one week before using for experiment. Wistar rats were fed either chow diet or high-phosphate diet in the absence or presence of 0.75% adenine to induce chronic kidney disease. Adenine was withdrawn from the high-phosphate diet after four weeks, and rats were also fed with high-phosphate diet for four weeks to development vascular calcification. OCA was administered to rats by gavage at a dose of 10 mg/kg/day. In order to detect the effect of OCA, one group of rats was given OCA in parallel with feeding high-adenine diet (named OCA group), while the other group was given OCA by four weeks after feeding high-adenine diet (named OCA-T group). To inhibit miR135a-5p, rats were injected with antagomiR135a-5p (80 μg/g) through vena caudalis for three consecutive days before inducing chronic kidney disease.

1.3% phosphorus, 1.06% calcium, 1000 IU/kg vitamin D3, and 23% protein) in the absence or presence of 0.75% adenine, as previously described, to induce CKD[56,57]. Adenine was withdrawn from the high-phosphate diet after four weeks. OCA prepared with 0.5% methylcellulose was administered to the rats at a dose of 10 mg/kg/day by gavage. In order to detect the effects of OCA, rats from one group were simultaneously administered with OCA and fed with the high-adenine diet (named the OCA group), while those in the other group were administered with OCA after four weeks of being fed with the high-adenine diet (named the OCA-T group). To inhibit miR-135a-5p, the rats were injected with antagomiR-135a-5p (Sequence 5′–3′: UCACAUAGGAAUAAAAAGCCAUA) (at a dose of 80 μg/g) through the vena caudalis for three consecutive days before the induction of CKD.

**Calcification analysis.** Calcium deposition in the HASMCs and rat aortas was measured using Alizarin Red S (ARS) staining and Von Kossa staining, as described previously[58]. The cells or aortas were gently washed and fixed with 4% formaldehyde for 15 min at room temperature. Alizarin Red S solution (A5533, Sigma, St. Louis, MO, USA) was added to each well or section, followed by incubation for 20 min. After removing the dye, the cells or aortas were inspected and photographed using a phase-contrast microscope. For Von Kossa staining, after being fixed in 4% paraformaldehyde for 24 h, the aortas were dehydrated, embedded, and sectioned into 8-μm-thick sections. In accordance with the Von Kossa Staining Kit (ab150687, Abcam, Cambridge, MA, USA) protocol, the sections were deparaffinized, dehydrated, and incubated with 5% AgNO₃ solution for 30 min under ultraviolet light. After being rinsed several times with distilled water (fresh distilled water for each rinse), the sections were incubated with sodium thiosulfate solution and then with nuclear fast red solution. Calcium-rich areas were identified based on the appearance of black dots. The extent of calcified lesions was expressed as the percentage of the total aortal surface area covered by lesions, and the mean value of the calcified lesion area in each aortic section was calculated as described previously[59].

**Calcium content assay.** The calcium contents in the HASMCs and aortas were quantified using a calcium detection assay kit (ab10205, Abcam, Cambridge, MA, USA). Sample preparation and assay procedures were performed in accordance with the instructions provided in the kit. The calcium contents were normalized to

the protein concentrations, which were determined using a BCA protein assay kit (Interchim, Montluçon, France).

**Intracellular NF-κB content assay**. The intracellular NF-κB contents were measured using the NF-κB p65 ELISA kit (ab176663, Abcam, Cambridge, MA, USA). Sample preparation and assay procedures were performed in accordance with the instructions provided in the kit. The total protein concentrations of the samples were in the range of 400–500 µg/ml. The ODs were recorded at a wavelength of 450 nm, and the NF-κB contents were normalized to the corresponding total protein concentrations of the samples.

**Real-time RT-PCR**. Total RNA was extracted from the HASMCs and rat tissues using TriZol (Invitrogen, CA, USA) and reverse transcribed to cDNA by using the High-Quality Reverse Transcription Reagents Kit (Applied Biosystems by Thermo Fisher Scientific). TaqMan RT-PCR was performed in triplicate using the ViiA™ 7 Real-Time PCR System (Applied Biosystems) with the following probes (Life Technologies): 478581_mir (hsa-miR-135a-5p/rno-miR-135a-5p), Hs00610320_m1 (human TGFBR1), Rn00688966_m1 (rat tgfbr1), Hs00751239_s1 (human Msx2), Hs01047973_m1 (human Runx2), Hs01866874_s1 (human Osterix/SP7), Hs01029144_m1 (human ALP), Hs00959010_m1 (human OPN/SPP1), Hs01587814_g1 (human BGLAP/OCN), Hs06633309_s1 (human NF-κB), Hs02621508_s1 (human TNF-α), Hs00174131_m1 (human IL-6), Hs00196143_m1 (human TAB1), Rn01504766_m1 (rat TAB1), Hs00355671_g1 (human NFKBIA/IκBα), Rn01473657_g1 (rat Nfkbia /IκBα), Hs00177373_m1 (human MAP3K7/TAK1), and Rn01437015_m1 (rat Map3k7/TAK1). The relative quantitation of gene expression was performed using the comparative $C_T$ ($\Delta\Delta C_T$) method. Normal mRNA assays were run in the fast mode, and the relative mRNA expression levels were normalized to the β-actin expression levels. Unlike the PCR assays for normal mRNAs, those for the miRNAs were run in the standard mode, and the relative miRNA expression levels were normalized to the U6 expression levels.

**Western blotting**. The HASMCs and rat tissues were lysed with RIPA buffer containing complete protease and phosphatase inhibitor cocktail (Thermo Fisher Scientific). Protein concentrations were measured using a BCA protein assay kit (Interchim, Montluçon, France). The samples were separated by 10% or 12% SDS-polyacrylamide gel electrophoresis (SDS-PAGE), and the resulting protein bands were transferred onto PVDF membranes. After being blocked, the membranes were incubated with the following primary antibodies overnight at 4 °C: NF-κB antibodies (1:1000, ab32536, Abcam), TNF-α antibodies (1:1000, ab6671, Abcam), FXR antibodies (1:500, sc-13063, Santa Cruz Biotechnology), TAK1 antibodies (1:1000, ab109526, Abcam), p-TAK1 antibodies (1:1000, ab109404, Abcam), TAB1 antibodies (1:1000, ab227210, Abcam), and Phospho-IκBα antibodies (Ser32/36) (1:1000, #9246, Cell Signaling Technology). Then, the membranes were incubated with horseradish peroxidase-conjugated secondary antibodies at room temperature for 2 hours. The protein bands were detected using an ECL Substrate (Thermo Fisher Scientific), and the quantification of the average densities of the protein bands was performed using the Image-Pro Plus 4.5 software.

**Immunofluorescence staining**. The cells were first grown on an 8-well slide; then, they were incubated with the NucBlue® Live Cell Stain reagent (Invitrogen) for 20 min, fixed with 4% paraformaldehyde for 15 min, and incubated with 0.1% Triton 100 for 20 min. After being blocked with 2% BSA in PBS for 20 min, the cells were incubated with the following primary antibodies overnight at 4 °C: NF-κB antibodies (1:100, ab32536, Abcam), TNF-α antibodies (1:100, ab6671, Abcam), TAK1 antibodies (1:1000, ab109526, Abcam), TAB1 antibodies (1:200, ab227210, Abcam), and p-IκBα (B-9) antibodies (1:100, sc-8404, Santa Cruz Biotechnology). The cells were then washed with PBS, incubated with the corresponding secondary antibodies for 1 hour, and visualized using a fluorescent microscope (DM16000B, Leica, Heerbrugg, Switzerland). The tissues were cut into 8-µm-thick sections. After dewaxing, dehydration, and antigen retrieval, the sections were blocked using 5% serum, incubated with the primary antibodies overnight at 4 °C as mentioned above, and then incubated with biotin-labeled secondary antibodies for 1 h at room temperature.

**Transfection**. The cells were seeded into 6-well plates at a density of $8 \times 10^5$ cells/well. Twenty-four hours later, they were transfected with a miR-135a-5p inhibitor (#4464084, Thermo Fisher Scientific) (or mimic (#4464066, Thermo Fisher Scientific)) and TGFBR1 siRNA (#AM51331, Thermo Fisher Scientific). According to the manufacturer's instructions, the amounts of Lipofectamine® RNAiMAX (Invitrogen) and miR-135a-5p inhibitor (or mimic) and TGFBR1 (or TAK1) siRNA to be added are 7.5 µL per well and 25 pmol per well, respectively, along with Opti-MEM® Medium (Gibco). The cells were first incubated with complex medium for 6 h; then, the complex medium was replaced with culture medium, followed by incubation for another 18 h. The cells were then subjected to the subsequent treatments.

**Micro-CT analysis**. The rats were anesthetized with isoflurane and scanned using the Quantum GX2 micro-CT Imaging System (PerkinElmer, USA) at a voxel size of 144 µm, and a voltage of 70 kV, with a current of 60 µA. Pictures of the chest areas (including heart and thoracic aorta) of the rats were taken; the processing of the images and three-dimensional standard microstructural analysis were performed using the Analyze 12.0 software (AnalyzeDirect).

**Blood chemistry and urine analysis**. The serum concentrations of creatinine, urea nitrogen, calcium, and phosphate were determined by using the Beckman Coulter AU5800 instrument. The concentrations of microalbumin in urine were tested using a Roche C702 system.

**Statistics and reproducibility**. Data from at least three independent experiments are presented as the mean ± SD. Comparison between data from two groups was performed using a t-test for independent samples, whereas one-way ANOVA with Bonferroni's multiple comparison post hoc test was used for the statistical analyses of data from multiple groups. P-values < 0.05 were considered statistically significant.

**Reporting summary**. Further information on research design is available in the Nature Research Reporting Summary linked to this article.

## Data availability
The data that support the findings of this study are available on request from the corresponding author (Y.L.L) upon reasonable request.

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

## Acknowledgements
We thank the members of the Department of Clinical Pharmacology and Toxicology at the University Hospital Zurich for their helpful comments on the paper. This study was supported by the National Natural Science Foundation of China (81774242), Natural Science Foundation of the Shandong Province (ZR2018ZC1157), Taishan Scholar Post Construction Fund (ts201712042) and Major Science and Technology Innovation Project in Shandong Province (2017CXGC1307).

## Author contributions
C.L and Z.B.G. conceived the project. C.L, S.J.Z., X.Q.C., J.K.J., and W.Q.Y. performed the experiments. Y.L.L. supervised the study and provided guidance. C.L. wrote the paper. All authors have reviewed the paper.

## Competing interests
The authors declare no competing interests.
