## [Peer Review File · Communications Biology]

Reviewers' comments:

Reviewer #1 (Remarks to the Author):

In this manuscript, the authors described the mechanism of vascular inflammation and calcification by using HASMCs and HAD treated rats. The results of the experiments led to several conclusions: 1) high phosphate medium causes HASMC inflammation by the activation of the TGFBR1/TAK1 signaling pathway, 2) FXA activation decreased the calcification levels by reducing TGFBR1/ TAK1-mediated inflammation, 3) miR-135a-5p is up-regulated by FXR activation and attenuates the vascular inflammation and calcification.

There are several points the authors should reconsider.

Major points:

Introduction:

In the last paragraph, it would be better to describe a bit about miR135a-5p more specifically.

Results:

- 1) In Fig. 1E, the Western blot data is not very clear to conclude that TGFBR1 silencing decreases the expression of proinflammatory cytokines.
- 2) The readers would like to know why vascular calcification occurred in HAD rats. It would be better to show the serum phosphate levels in HAD and HPD groups.
- 3) In Fig. 2F, is there a significant difference in NF-kB TNFa between HPD and HAD?
- 4) In Fig. 2G, data for HPD are missing.

Discussion:

- 1) In HASMC inflammation, how is TGF- β 1 or TGFBR2 activated by high phosphate exposure.
- 2) miR-135a-5p may be a key target for vascular calcification, but it could not reverse calcified areas that have been calcified according to the CKD rat experiment. The authors should describe somewhere in Discussion.

Minor points:

- 1) Line 151, what is OCA? You should not abbreviate when it first appears in the sentence. How did you determine the optimal concentration.
- 3) Figure legends for Fig. 2E and F are wrong.
- 4) Line 241, miR-135-5p inhibitor was changed to antagomiR-135a-5p without any description.
- 5) In Fig. 7 legend, (80 μ g/g) should be (80 μ g/g body weight).
- 6) Line 338, what is BA?

Reviewer #2 (Remarks to the Author):

The manuscript demonstrated the role of TGFBR1/TAK1 in vascular inflammation and calcification. They also claimed that FXR-mediated miR-135a-5p expression inhibited TGFBR1-mediated pathogenic observations. The present study is interesting; however, there are critical concerns for publication, especially in TAK1 activation.

Major concerns

1. The most critical concern is why the authors focused on non-canonical TAK1 pathway, but not canonical Smad pathway. It is essential to characterize the role of Smad in all experiments performed in this study and discuss why TAK1 pathway is important for the observations.
2. The protein expression level of TAK1 and TAB1 was analyzed. There is no direct evidence whether the protein expression correlated with activation status of TAK1. It essential to demonstrate that TAK1 is activated by testing the phosphorylation status of TAK1.
3. All Western blot analyses are not clear, especially the effects of siRNA (Fig. 1E), OCA (Fig. 3B) and miR mimic (Fig. 6C). In Fig. 2G, because actin expression was also increased, is the increase in TAK1 expression significant?

4. TAK1 is known to be an upstream kinase of NF- κ B; however, it is essential to examine its role in HASMCs. RNAi knockdown is better to characterize the role of TAK1.
5. Overall, information of stimulation time in the experiments using cultured cells are limited. All the experiments show endpoint only. Time course during 7 days is important factors for evaluating the role TAK1 and NF- κ B. TNF- α is known as a strong activator of TAK1. There is a possibility that TNF secreted by Pi+Ca stimulation triggered NF- κ B activation.
6. TAK1 is a MAP3K that is involved in JNK and p38 mainly. Are there any roles of MAPKs in vascular inflammation and calcification?

Other points

1. The order of supplemental Figures 1 and 2 is not correct (lines 99 and 102).
2. Why kidney size was reduced by OCA treatment (Supplemental Figure 3F).

Dear Editor and Reviewers,

Thank you very much for your letter and for the reviewer's comments concerning our manuscript entitled "Farnesoid X receptor activation inhibits TGFBR1/TAK1-mediated vascular inflammation and calcification via miR-135a-5p". We greatly appreciated for your helpful and insightful comments and critiques. We have modified the manuscript accordingly and have edited figure and supplementary figure as suggest in the revision, which we hope to meet your request. We believe that the quality of revised manuscript has been significantly enhanced based on reviewer's comments. All changes made to the text are highlighted for your easy identification.

With regard to the reviewer(s)' comments and suggestions, we wish to reply as follows:

Reviewer #1 (Remarks to the Author):

In this manuscript, the authors described the mechanism of vascular inflammation and calcification by using HASMCs and HAD treated rats. The results of the experiments led to several conclusions: 1) high phosphate medium causes HASMC inflammation by the activation of the TGFBR1/TAK1 signaling pathway, 2) FXA activation decreased the calcification levels by reducing TGFBR1/ TAK1-mediated inflammation, 3) miR-135a-5p is up-regulated by FXR activation and attenuates the vascular inflammation and calcification.

There are several points the authors should reconsider.

Major points:

Introduction:

In the last paragraph, it would be better to describe a bit about miR135a-5p more specifically.

*Response: Thanks for your suggestion. We have added a little bit introduction to miR-135a-5p in **line 79** of the revised manuscript. As "MiR-135a-5p has been demonstrated to be involved in a variety of cellular physiological processes. It was reported to show an inhibitory role in several types of cancer including thyroid carcinoma and glioblastoma^{30,31}. In addition, miR-135a-5p participates in the*

neuroprotective effect of hydrogen sulfide on Parkinson's disease³²." For the role of miR135a-5p in vascular calcification, we describe it in discussion combination with present data.

Results:

1) In Fig. 1E, the Western blot data is not very clear to conclude that TGFBR1 silencing decreases the expression of proinflammatory cytokines.

Response: Thank you for your careful work. We put these data in the figures because the gray value differences of bands is statistically significant. However, we should choose the clearer picture to show. We are sorry for that and provide some clearer data in revised manuscript according to your suggestion.

2) The readers would like to know why vascular calcification occurred in HAD rats. It would be better to show the serum phosphate levels in HAD and HPD groups.

*Response: Thank you for your suggestion. We have added the levels of serum creatinine, serum phosphate, and urea nitrogen in the supplemental figure 4. The result showed that high levels of serum creatinine, urea nitrogen, and phosphate were observed in rats fed with the high-adenine diet (HAD group), indicating the impairment of renal function; however, this was not the case in rats fed with a high-phosphate diet (HPD). We also have modified it in the **line138** of the revised manuscript.*

3) In Fig. 2F, is there a significant difference in NF- κ B TNF α between HPD and HAD?

Response: Thanks for your careful review. High-phosphate diet (HPD) can't induce chronic kidney disease and vascular calcification. High-adenine diet (HAD) is based on HPD plus adenine to induce chronic kidney disease and vascular calcification. Therefore, HPD was established as a dietary control group of HAD. There was indeed a significant difference in NF- κ B and TNF- α expression between HPD and HAD groups. The results showed that there was a significant inflammatory response in HAD-induced vascular calcification. HPD didn't induce vascular calcification, and there was no obvious inflammatory response.

4) In Fig. 2G, data for HPD are missing.

Response: Thanks for your careful review. Because HPD group was just used as the control group of HAD group and the results showed no vascular calcification and inflammation in HPD group. We aimed to explore the mechanism of vascular calcification and inflammation induced by HAD. Therefore, the upstream activation mechanism of vascular inflammation and calcification in HPD group is not necessary. So we didn't measure the expression of TAK1, TAB1 and p-I κ B α in HPD group.

Discussion:

1) In HASMC inflammation, how is TGF- β 1 or TGFBR2 activated by high phosphate exposure.

*Response: Thanks for your review. Studies have demonstrated that TGF- β 1 expression is increased by phosphate in VSMCs^{37,38}. TGF- β 1 is the ligand of TGFBR2, and after TGF- β 1 binds to TGFBR2, TGFBR1 is recruited and phosphorylated^{18,19}. We have supplemented these discussion in **line 325** of the revised manuscript according to your advice.*

2) miR-135a-5p may be a key target for vascular calcification, but it could not reverse calcified areas that have been calcified according to the CKD rat experiment. The authors should describe somewhere in Discussion.

*Response: Thanks for your review. We have added these discussion in **line 396** of the revised manuscript according to your suggestion. AS “Therefore, miR-135a-5p may be also a key target for vascular calcification, since increased miR-135a-5p expression by FXR activation reduces the formation of vascular calcification. However, miR-135a-5p has no effect when vascular calcification has been formed, so it could not reverse calcified areas that have been calcified according to the CKD rat experiment.”*

Minor points:

1) Line 151, what is OCA? You should not abbreviate when it first appears in the sentence. How did you determine the optimal concentration?

Response: I'm sorry we neglected the full name of OCA, OCA is the abbreviation of obeticholic acid. We have corrected in revised manuscript as “obeticholic acid (OCA, a FXR agonist)”. We measured the intracellular calcium content after the intervention

of 1, 3 and 5 μ M OCA. We have found that there were similar effect on 3 and 5 μ M, better than 1 μ M. Therefore, we chose 3 μ M as the intervention concentration, consistent with previous studies (Miyazaki-Anzai, S. et al. Farnesoid X receptor activation prevents the development of vascular calcification in ApoE^{-/-} mice with chronic kidney disease. *Circulation research* 106, 1807-1817, (2010).).

2) Figure legends for Fig. 2E and F are wrong.

Response: Thanks for your careful review. We are very sorry for this mistake and we have modified it in revised manuscript.

3) Line 241, miR-135-5p inhibitor was changed to antagomiR-135a-5p without any description.

*Response: Thanks for your concern. We have added an explanation of why antagomir-135a-5p was used in **line 253** of the revised manuscript according to your suggestion. As “Since antagomirs specific and complementary to their mature target miRNA, interfere with its function and are proved to be efficient in many tissues, we used antagomiR-135a-5p to inhibit miR-135a-5p in animal experiments.”*

4) In Fig. 7 legend, (80 \square g/g) should be (80 \square g/g body weight).

Response: Thank you for your careful work. We have corrected it in revised manuscript according to your comment.

5) Line 338, what is BA?

Response: Thanks for your review. We changed BA to the full spelling “bile acid”.

Dear Editor and Reviewers,

Thank you very much for your letter and for the reviewer's comments concerning our manuscript entitled "Farnesoid X receptor activation inhibits TGFBR1/TAK1-mediated vascular inflammation and calcification via miR-135a-5p". We greatly appreciated for your helpful and insightful comments and critiques. We repeated and supplied the experiment that you suggested in the past four months. We have modified the manuscript accordingly and have edited figure and supplementary figure as suggest in the revision, which we hope to meet your request. We believe that the quality of revised manuscript has been significantly enhanced based on reviewer's comments. All changes made to the text are highlighted for your easy identification.

With regard to the reviewer(s)' comments and suggestions, we wish to reply as follows:

Reviewer #2 (Remarks to the Author):

The manuscript demonstrated the role of TGFBR1/TAK1 in vascular inflammation and calcification. They also claimed that FXR-mediated miR-135a-5p expression inhibited TGFBR1-mediated pathogenic observations. The present study is interesting; however, there are critical concerns for publication, especially in TAK1 activation.

Major concerns

1. The most critical concern is why the authors focused on non-canonical TAK1 pathway, but not canonical Smad pathway. It is essential to characterize the role of Smad in all experiments performed in this study and discuss why TAK1 pathway is important for the observations.

Response: Thanks for your review. We have established a microRNA database of vascular calcification in previous studies. It was showed that TGF- β pathway played a key role in vascular calcification through enrichment analysis. Then, we measured the expression of the downstream of TGF- β pathway in osteogenic medium-induced HASMCs calcification, and BMP2, Smad3, Smad2, TAK1 and TAB1 were also

increased. Several studies have reported that the TGF- β /Smad and BMP2/Smad pathways are directly involved in vascular calcification^{33,34}. However, to date, no evidence has demonstrated whether TAK1 is a key potential target of osteogenic medium-induced HASMC calcification. Therefore, in this study, we focused on non-canonical TAK1 pathway and revealed that the TGFBR1/TAK1 pathway-mediated HASMC inflammation was involved in the pathological process of vascular calcification.

2. The protein expression level of TAK1 and TAB1 was analyzed. There is no direct evidence whether the protein expression correlated with activation status of TAK1. It is essential to demonstrate that TAK1 is activated by testing the phosphorylation status of TAK1.

Response: Thanks for your suggestion. The consideration raised by the reviewer is a solid concern. We repeatedly detected the phosphorylation level of TAK1 in all the involved experiments over the past four months, and the results showed an increased expression of p-TAK1. We have provided the Western blot analyses of p-TAK1 in all revised figures.

3. All Western blot analyses are not clear, especially the effects of siRNA (Fig. 1E), OCA (Fig. 3B) and miR mimic (Fig. 6C). In Fig. 2G, because actin expression was also increased, is the increase in TAK1 expression significant?

Response: Thank you for your careful work. We put these data in the figures because the gray value differences of bands is statistically significant. However, we should choose the clearer picture to show. We are sorry for that and provide some clearer data according to your suggestion. In Fig. 2G, the increase in p-TAK1 expression is indeed significant, and we showed the better data.

4. TAK1 is known to be an upstream kinase of NF- κ B; however, it is essential to examine its role in HASMCs. RNAi knockdown is better to characterize the role of TAK1.

Response: Thank you for your valuable advice. According to your suggestion, we examined the role of TAK1 in osteogenic medium-cultured HASMCs. As showed in supplemental figure 5, TAK1 knockdown reduced the expression of NF- κ B and TNF- α ,

*which induced by osteogenic medium. TAK1 knockdown also reduced increased amount of NF- κ B and calcium in cells, and attenuated osteogenic medium-induced HASMCs calcification. These data suggest that that TAK1 plays a crucial role in osteogenic medium-induced HASMCs inflammation and calcification. We have supplemented these results in **line 128** of the revised manuscript.*

5. Overall, information of stimulation time in the experiments using cultured cells are limited. All the experiments show endpoint only. Time course during 7 days is important factors for evaluating the role TAK1 and NF- κ B. TNF- α is known as a strong activator of TAK1. There is a possibility that TNF secreted by Pi+Ca stimulation triggered NF- κ B activation.

*Response: Thank you for your valuable and thoughtful comments. According to your suggestion, we measured the expression of p-TAK1 and NF- κ B in the cells of each day after treating with osteogenic medium. At the same time, we also analyzed the calcium content in cells. As shown in supplemental figure 6, the results showed that the expression of p-TAK1 and NF- κ B increased significantly after three days of treating with osteogenic medium, while the calcium content in cells increased significantly after the fifth day ($p < 0.01$) of intervention. We have supplemented these results in **line 116** of the revised manuscript.*

We agree with the effect of TNF secretion on NF- κ B activation. But after silencing the expression of TGFBR1, the activation of p-TAK1 and NF- κ B were reduced in osteogenic medium-cultured HASMCs, and the expression of TNF- α was also decreased. The data of miR-135a-5p and FXR activation also supplied the evidence that the TGFBR1/TAK1 pathway-mediated HASMC inflammation was involved in the pathological process of vascular calcification.

6. TAK1 is a MAP3K that is involved in JNK and p38 mainly. Are there any roles of MAPKs in vascular inflammation and calcification?

Response: Thanks for your concern. This is a very important and meaningful reflection. We also thought about whether MAPKs was involved in vascular inflammation and calcification. However, some studies have demonstrated that p38 MAPK activation is involved in the Pi-induced smooth muscle cells inflammation and

*calcification, and inhibition of p38 MAPK decreases the smooth muscle cells calcification⁴¹⁻⁴³. Therefore, we didn't focus on MAPKs pathways. These studies also provide evidences that TGFBR1/TAK1 could be a good target for reducing vascular calcification. We have added these discussion in in **line 336** of the revised manuscript.*

Other points

1. The order of supplemental Figures 1 and 2 is not correct (lines 99 and 102).

Response: Thanks for your careful review. We have corrected the order of supplemental Figures 1 and 2 in revised manuscript.

2. Why kidney size was reduced by OCA treatment (Supplemental Figure 3F).

Response: We made a statistical analysis of all kidney sizes, and the results showed that there was no significant differences between HAD group and HAD+OCA group. Since the size of the kidney is not directly related to this study, we decided to delete the picture after careful consideration. We also corrected it in revised manuscript.

REVIEWERS' COMMENTS:

Reviewer #1 (Remarks to the Author):

The authors have appropriately revised the manuscript.

Minor: The figure legend of the schematic model in Figure 8 should be described in the present tense.

Reviewer #2 (Remarks to the Author):

The authors responded to my original comments carefully; however, I have one remaining point for this manuscript.

1. Based on the comment 2, they newly performed the experiments to replace TAK1 blots with pTAK1 blots. I appreciated the efforts, which clearly demonstrated the activation of TAK1. However, it is better to show both total TAK1 and pTAK1 blots, because TAK1 activation is largely dependent on the increased expression of TAK1 protein.